# A Novel Mechanically Robust and Biodegradable Egg White Hydrogel Membrane by Combined Unidirectional Nanopore Dehydration and Annealing

**DOI:** 10.3390/ijms241612661

**Published:** 2023-08-10

**Authors:** Xuan Dong, Yu-Qing Zhang

**Affiliations:** 1School of Biology and Basic Medical Sciences, Medical College, Soochow University, RM702-2303, No. 199, Renai Road, Industrial Park, Suzhou 215123, China; 20214009026@stu.suda.edu.cn; 2College of Chemistry, Chemical Engineering and Materials Science, Soochow University, RM702-2303, No. 199, Renai Road, Industrial Park, Suzhou 215123, China

**Keywords:** egg white, unidirectional nanopore dehydration, annealing, hydrogel, mechanical properties, biodegradation

## Abstract

A homogeneous egg white obtained by high-speed shearing and centrifugation was dehydrated into a fragile and water-soluble egg white glass (EWG) by unidirectional nanopore dehydration (UND). After EWG annealing, it can become an egg white hydrogel membrane (EWHM) that is water-insoluble, flexible, biocompatible, and mechanically robust. Its tensile strength, elongation at break, and the swelling ratio are about 5.84 MPa, 50–110%, and 60–130%, respectively. Protein structure analysis showed that UND caused the rearrangement of the protein molecules to form EWG with random coil and α-helix structures. The thermal decomposition temperature of the EWG was 309.25 °C. After EWG annealing at over 100 or 110 °C for 1.0 h or 45 min, the porous network EWHM was mainly composed of β-sheet structures, and the thermal decomposition temperature increased to 317.25–318.43 °C. Their 12-day residues in five proteases ranged from 1% to 99%, and the order was pepsin > neutral protease > papain > trypsin > alkaline protease. Mouse fibroblast L929 cells can adhere, grow, and proliferate well on these EWHMs. Therefore, the combined technology of UND and annealing for green and novel processing of EWHM has potential applications in the field of biomimetic and biomedical materials.

## 1. Introduction

Egg white (EW) is a natural and viscous aqueous solution composed of many proteins and other substances and is a nutritious food with a long history. It is an abundant, cheap, and natural source of important proteins, such as ovalbumin and lysozyme, and is a new ancient biomaterial [1]. Because EW is a highly functional food protein, it is often mixed into food matrixes and has excellent biocompatibility, thermal gelation, biodegradability, and bio-absorption [2]. With a variety of physiological functions and biochemical activities, it is closest to the human body in amino acid composition, plays an important role in maintaining human health, and has been widely used in medical, food, bioengineering, and other fields. In addition to the wide application to food science mentioned above [3], recently EW has attracted great attention in the fields of edible packaging materials [4], cosmetics [5], biomedicine [6], sustained-release drug carriers [7], biomedicine [8], energy [9], textile materials [10], optoelectronic materials [11], and biomaterials [12]. Recently, the author also wrote a detailed review of the current research on EW [13].

EW from chicken eggs, which accounts for about 60% of the total weight of an egg, is a yellowish, transparent, and sticky colloidal solution composed of 11% proteins and 88% water. The proteins of EW mainly consist of five kinds of glycoproteins: ovalbumin (54%), ovotransferrin (13%), ovomucoid (11%), ovomucin (3.5%), and lysozyme (3.4%) [14]. Because of its high nutritional value, EW has become an important source of edible membranes; it is expected to replace conventional plastics and reduce pollution. In order to overcome the shortcomings of EW films, which include poor mechanical, gelation, and solubility properties, chemical cross-linking is often performed to develop novel edible membranes, making use of the numerous disulfide bonds and sulfhydryl groups of EW [15,16]. Edible packaging films with certain mechanical properties and oxygen barriers have been developed by adding edible plasticizers, such as glycerol [4,17], chemical or biological cross-linking agents [18], or by adding lipid films [19], gelatin [20], or casein bioplastic films [21].

EW has been used to encapsulate and coat curcumin capsules, titanium dioxide nanoparticles, and biological nanocomposite hydrogels and nanoparticles for the sustained release and delivery of antineoplastic drugs [6,22,23,24,25]. The EW–bovine gelatin composite hydrogels by ultrasound and transglutaminase pretreatment has potential uses in various fields, primarily in terms of controlled delivery in the food and pharmaceutical industries [26]. After the addition of polyvinyl alcohol, the EW mixture has also been used as a wound-healing dressing [27]; when hydroxyapatite, fish collagen, and alumina paste is added, EW has been synthesized into foamed EW bioceramics with high water absorption, suitable for bone growth [12,28,29,30]. EW sponges prepared by cross-linking or modification have been used as a new type of soft-tissue scaffold [31]. A bionic membrane with a melanin biosynthesis function has been prepared by cross-linking with dihydroxyindole [32] and mixed with keratin to form a luminescent gold-cluster composite film [33]. EW has also been used as a green resist for micro- and nanoscale patterns in lithography [34] and mixed with nitric acid metal salts and calcined at a high temperature to form single-phase cubic ferrite with good magnetic properties [35]. EW has also been used as a substitute for dielectric materials when preparing field effect transistors [11] and highly ordered inorganic/organic hybrid materials [36]. 

The EW biomaterials discussed above are usually prepared by adding other substances, including chemical cross-linking agents. There have only been few studies of the preparation of EW biofilms with excellent mechanical properties using only EW. Recently, the authors have just reported a mechanically robust egg white hydrogel scaffold with excellent biocompatibility by three-step green processing [37]. In this paper, completely green processing technology has been developed, using no chemical reagents in the preparation process. Homogenized EW is processed into an egg white hydrogel membrane (EWHM) by unidirectional nanopore dehydration (UND), leading to molecular rearrangement and secondary structure changes in the protein.

## 2. Results

### 2.1. Preparation of EWG and the Effect of Temperature and Humidity

After adding 5.0 mL homogenized EW liquid through the top hole (Figure 1a), the rubber plug was inserted to seal the internal space, and the mold containing EW was placed horizontally on the mold rack. Under certain temperatures and a relative humidity (Figure 1b) box, the EW liquid was dehydrated unidirectionally through the nanopore filter for 8–96 h to form EWG. Figure 1b reports the time required for crystallization (h) under different temperatures (in the range 5–55 °C) and humidity (in the range 10–95% RH). “**×**” is used to indicate that an intact EWG cannot be formed under these conditions. 

The experimental results showed that intact EWGs could be formed from 5 °C to 55 °C, and the time of formation decreased with increased temperature or decreased humidity. The time required to form EWG at 5 °C and high humidity is very long, about 4 days. Complete EWGs can be formed after dehydration for 18 h at lower humidity (10%). At 45 °C and low humidity (10% RH), dehydration to form EWG required only 8 h. When the temperature is too high (55 °C) and the humidity is too high (>80% RH) or too low (10% RH), it is disadvantageous for the EW liquid to form qualified EWG. Therefore, the shaded part in the middle of Figure 1b is the most suitable temperature and humidity for the preparation of EWG. That is to say, the time range for preparing the best performing EWG under conditions of 15–35 °C and 65–35% RH is between 20 h and 54 h (green area in Figure 1b). Under this optimal condition, five UND-based EWGs were obtained from five eggs, (a) ostrich, (b) goose, (c) duck, (d) chicken, and (e) quail eggs (see Appendix A). In fact, the formation time of EWG is not only related to temperature and humidity, but it is also closely related to the airflow under the mold and the pore size of the filter membrane. 

### 2.2. Effect of Annealing Mode on Mechanical Properties of EWHM

In order to improve the poor mechanical properties and water solubility of the EWG, three main annealing methods were tested: boiling, steaming, and oven heating at a minimum temperature of 100 °C. In the boiling test, the EWG was directly immersed in boiling water for heat treatment. As shown in Figure 2a,b, when EWG was directly placed in boiling water, its tensile strength decreased gradually with boiling time, and the tensile strength and elongation at break were best following 5 min of boiling, but the EWHM surface appeared frosted after boiling, indicating that direct boiling of EWG leads to a milky-white surface, poor transparency, and other undesirable phenomena (Figure 2c). These shortcomings worsened with extended boiling time, and the surface color continued to darken. Second, when EWG was directly steamed for 5–90 min, the tensile strength first increased, then decreased. Steaming the EWG for 20–30 min attained the maximum tensile strength, that is, the tensile strength when steamed for 20 min reached 17.10 MPa, but the elongation remained similar to that of the boiled samples. Finally, oven heating at 100 °C was not as effective as the boiling and steaming heat-treatment methods. Oven heating required 30 min to achieve the maximum tensile strength at 100 °C, but the tensile strength was still lower than that of the other two heat treatments. The maximum tensile strength was only 1/2–1/3 of the other two groups. These results showed that the method of heat treatment has a significant effect on the surface morphology and mechanical properties of the EWHM. Different heat-treatment methods cause differences in tensile strength and tensile properties of the EWHM, which may be mainly related to the exposure environment of EWG, especially the amount of water, thus affecting the speed of heat transfer and changes in protein structure.

### 2.3. Effects of Alternating Water Boiling and Watertight Boiling on Mechanical Properties and Transparency of EWHM

In order to improve the mechanical properties and ensure the transparency of the EWHM, alternating treatments of boiling and watertight boiling were designed as a post-treatment for the EWHM in this experiment. The homogenized EW liquid (5.0 mL) was added to the custom mold, placed at 25 °C and 50% RH under a micro-fan blast, and EWG was formed by UND for 12–24 h. It was then moved into an airtight, waterproof polypropylene (PP) pocket and placed in a boiling water bath for two kinds of heat treatment according to the times shown in Figure 3a. One heat treatment was to watertight boil the sample for 5–60 min and then boil the sample in water for 10 min. Another heat treatment was to watertight boil the sample in the PP pocket for 5 min, then boil it in water for 5–60 min (Figure 3b). After treatment, according to the time shown in Figure 3, the EWG becomes a water-insoluble and almost transparent EWHM. The EWHMs were immersed in 37 °C water for 24 h prior to determining the tensile properties. Then, the mechanical properties were tested on a tensile machine. Each sample was cut into the rectangular strips (6.0 mm × 25 mm), and 10 measurements were repeated to calculate the average values (±SD).

Figure 3a shows that the tensile force reached the peak value of 11.72 N, and the elongation at break reached 120.16% after 20 min of boiling, followed by 10 min of watertight boiling. By watertight boiling for 5 min and then boiling the sample for 5–60 min (Figure 3b), the mechanical properties of samples watertight boiled for 5 min followed by boiling for 20 min were the best, with a tensile force of 13.69 N and an elongation at break of 222.52%. However, the surface of the EWHM frosted with extended boiling time (asterisked in Figure 3b), but not in the other seven groups. Therefore, all subsequent post-treatment experiments use 10 min of watertight boiling as a pre-treatment to ensure transparency, followed by other heat treatments using different methods and at different temperatures.

### 2.4. Effect of High-Temperature Oven-Heating Treatment on Tensile and Swelling Properties

In order to ensure the transparency of the EWHMs, all the EWG samples were pre-treated with 10 min of watertight boiling. The pre-heated samples were wrapped in aluminum foil to avoid direct exposure to high temperatures, and they were heated for 10 min at six high temperatures between 100 °C and 150 °C. Watertight boiling followed by oven heating at high temperatures could affect the tensile and swelling properties of the EWHM (Figure 4). Before the tensile properties were measured, the EWHM was immersed in 37 °C water for 24 h. The mechanical properties were tested on the universal material testing machine. Each sample was cut into 6.0 mm × 25 mm rectangular strips, and ten items were measured repeatedly to calculate the average and standard deviation. 

According to the experimental results shown in Figure 4a, the EWG was pre-treated with watertight boiling for 10 min and then heated individually in an oven from 100 °C to 150 °C for 10 min. The tensile properties of the EWHM were enhanced with increased temperature, reaching 13 N at 130 °C and 16.73 N at 150 °C. Interestingly, the increase in the temperature of the process leads to a reduction in the elongation at break. The swelling ratio is only 41.3% at the highest oven-heating temperature (150 °C) and reached the highest value of 73.3% at 120 °C. The diameter of the swollen EWHM was similarly affected. Therefore, the oven-heating temperature should not be too high; otherwise, the swelling ratio will be low, and the EWHM will become brittle and dark (Figure 4b).

### 2.5. Effect of Oven-Heating Temperature and Time on Tensile Properties

As mentioned earlier, all EWG samples were pre-treated with watertight boiling for 10 min to maintain their transparency. Those samples were wrapped in tin foil and then oven-heated for 10–90 min at 100, 110, 120, and 130 °C. The effect of this watertight boiling followed by high-temperature oven heating on the tensile and swelling properties of the EWHM is shown in Figure 5a,b. The tensile strength of the sample treated by oven heating at 100 °C increased gradually with increased treatment time, and the maximum tensile force was 17.17 N at 60 min, while the elongation at break was 90%, and its mechanical properties decreased with increased temperature. The thermal oven-heated treatment at 110 °C had a similar trend: the treatment temperature was raised by 10 °C, and the tensile strength continued to rise from an initial high value, peaking at 24.85 N after 45 min, and the elongation percentage was 111.1%. Subsequently, with the delay in the treatment time, the mechanical properties of the samples declined. There was a similar trend in the two highest-temperature treatments. The treatment at 120 °C took 30 min to reach the maximum value, while at 130 °C, the time to reach the maximum value was even shorter, i.e., only 20 min. Overall, with increased oven-heating temperature, the time for the tensile strength to reach the peak value gradually shortened. From the analysis of treatment temperature, the tensile strength of the EWHM treated at 110°C was the largest and then decreased gradually with increased heating temperature.

### 2.6. Optical Properties of EWG and EWHM

Under the conditions of 25 °C, 55% RH, and fan ventilation, using a nanofiltration membrane or dialysis membrane with an interception pore size of ~10 kDa, 5.0 mL of fresh EW liquid was dehydrated by gravity for 12–48 h to prepare 7 cm^2^ of EWG with a thickness of 0.8 mm. The formed EWG was immediately boiled for 10 min, watertight boiled for 10 min, and oven-heated at 100 °C for 30 min to obtain three kinds of EWHM, which were photographed (Figure 6). Figure 6a shows that the EWG is very transparent, like glass, but fragile and easily soluble in water. After 10 min of boiling in a PP sealed bag, the sample became an elastic, water-insoluble EWHM with good mechanical properties, but the transparency was poor (Figure 6b). When the EWG was placed in a PP sealed bag and watertight boiled for 10 min, the resulting EWHM remained transparent, became water-insoluble, and had good mechanical properties (Figure 6c). Finally, the EWHM obtained after heating at 100 °C for 30 min had stronger mechanical properties, but the color was darker (Figure 6d). In addition, the color change and color depth of the EWG and the EWHM formed after this kind of heat treatment is related to the heat-treatment methods and conditions; more importantly, it is caused by the Maillard reaction in EWG or the EWHM [38]. The homogenized EW contains a certain amount of reducing sugar, such as glucose, because this experiment has not been desweetened. At room temperature, the Maillard reaction between the reducing sugar and free amino acid residues in EWG and EWHM protein still occurs slowly, which causes EWG and EWHM to be yellow or light brown. 

### 2.7. Stress and Strain Curves of EWHMs

At 25 °C and 55% RH, with fan ventilation, using a nanofiltration membrane or dialysis membrane within interception pore size of ~10 kDa, 5.0 mL of fresh EW liquid was dehydrated in gravity for 12–48 h to prepare 50 cm^2^ of EWG, with a thickness of 0.8 mm. After pre-treatment with watertight boiling for 10 min, two kinds of EWHM were obtained by airtight oven heating at 100 °C for 1 h and airtight oven heating at 110 °C for 45 min, respectively, EWHMa and EWHMb. Before the tensile test, the pre-cut 6 × 30 × 0.80 mm samples were allowed to swell in water at 37 °C for 24 h, and the mechanical properties were tested using a material testing machine, and ten replicants were measured to calculate the average values (±SD). The stress and strain curves of two EWHMs were obtained from the EWG shown in Figure 6f. Figure 6e shows the glass-like yellowish EWG formed by unidirectional dehydration of homogenized EW liquid through nanopores, which was both transparent and water-soluble. If placed in drier conditions, such as ≥25 °C and ≤35% RH, the EWG spontaneously cracked and broke: it was so brittle that it broke without touching it. This fragile EW crystal can be transformed into an EW biomaterial with mechanical properties, especially strong tensile strength and elasticity (also see Appendix A), after heat treatment, such as boiling or steaming. As shown in Figure 6f, the EWHMa formed from pre-watertight-boiled EWG annealed at 100 °C for 60 min has a tensile strength of 3.4 MPa and an elongation ratio of 90%. After annealing at a higher temperature (110 °C/45 min), the tensile strength of EWHMb increased by 50% to 5.2 MPa. With the increase in treatment temperature, the tensile strength of EWHM improved, but the material became brittle and the color darkened, and the elongation ratio decreased with increased temperature.

### 2.8. SEM of the Cross-Section of EWHMs

From the above experimental results, it can be known that a brittle water-soluble EWG is formed by unidirectional dehydration of homogeneous EWs through nanopores. After a magnification of 150×, SEM observation shows that there are many vertical and horizontal cracks and irregular cracks on the surface (Figure 7a). After being annealed at a high temperature, no cracks were observed on the surface after magnification of 1.500 times (Figure 7b). Figure 7c shows the SEM photo of the longitudinal section of the EWHMa membrane at a magnification of 200×. It can be observed that the porous structure similar to bamboo knots is arranged in the longitudinal direction. When magnified to 2000×, a porous network structure with pore sizes ranging from 2 to 12 μm can be observed (Figure 7d). Therefore, the annealed EWHM formed by this UND is an ordered porous network structure, and it can be speculated that this hydrogel is very suitable for cell growth, proliferation, and implantation in vivo.

### 2.9. Thermal Analysis

Figure 8 shows the DSC patterns of EWG and the two kinds of EWHM. The thermal decomposition temperature of the water-soluble EWG was 314.7 °C, and that of the EWHMs was significantly greater. From Figure 8a, the EWG formed by UND was initially endothermic at 52.9 °C and its structure changed from random coil to α-helix, reaching a maximum of 77.8 °C. However, the two kinds of high-temperature-treated hydrogels, EWHMa and EWHMb, continued to be exothermic, less of their protein structure was random coil, and part of the α-helix structure changed directly to β-sheet. The EWG was initially exothermic from 77.8 °C, and the α-helix structures began to change to β-sheets, reaching an exothermic peak at 203.0 °C. With increasing temperature, it became endothermic, and the glass point transition temperature was 231.9 °C. The degradation of EWG began at 296.5 °C and peaked at 314.7 °C. However, the exothermic and endothermic processes of the two EWHMs were not as significant as those of EWG, and the maximum degradation peaks of EWHMa and EWHMb were observed at 318.7 °C and 320.1 °C, respectively. These results showed that after heat treatment, the structure of EWG changed from random coil and α-helix to a structure dominated by β-sheets, and their thermal stability was significantly higher than that of EWG. This result is basically consistent with the EW hydrogel scaffold prepared by three-step green method reported recently by the authors [37].

### 2.10. FTIR Spectra

Figure 8b shows that the amide I band of water-soluble EWG was at 1633.4 cm^−1^, and the amide II band was centered at 1536.5 cm^−1^, and a small shoulder peak at 1516.3 cm^−1^ was not very obvious, which indicates the dominance of random coil and partially α-helical structures. When the EWG was annealed with airtight baking at 100 °C for 60 min, the two bands of EWHMa shifted: the amide I and amide II peaks shifted to 1626.2 cm^−1^, and 1527.3 cm^−1^, respectively, and the small shoulder peak, which was not obvious before, became a very obvious peak at 1516.3 cm^−1^, which was due to the transition from the random crimp and α-helix structures to β-sheet structures. When the EWG annealed with airtight baking at 110 °C for 45 min became EWHMb, the amide I band shifted down to 1624.3 cm^−1^, the amide II band was similar to that of EWHMa, and two enhanced peaks still appeared at 1527.3 cm^−1^ and 1516.3 cm^−1^. These results show that annealing at higher temperatures transformed the α-helix structure to the β-sheet structure. 

### 2.11. XRD Spectra

From Figure 8c, it is obvious that the EWG with good water solubility had a diffraction peak at 2*θ* = 19.51°. This is different from the typical amorphous diffraction pattern of a soluble protein, that is, there is a broad peak in the range of 2*θ* from 5° to 50°, which contains the diffraction characteristics of α-helix crystals. There is also a diffraction peak at 2*θ* = 7.68°, indicating that UND promotes the formation of α-helix structures, which agrees with the FTIR spectroscopy results. When the transparent and almost water-soluble EWG was heat-treated at a high temperature, its diffraction pattern changed. After annealing at 100 °C for 60 min, the EWG was changed into a soft and elastic EWHM with strong mechanical properties in the wet state. The main peak appeared at 2*θ* = 21.04°, and small diffraction peaks also appeared at 2*θ* = 29.08° and 39.74°. When the heat-treatment temperature was raised to 110°C for 45 min, these diffraction peaks increased, appearing at 2*θ* = 30.51°, and a small diffraction peak at 2*θ* = 39.74° appeared. Its mechanical properties, especially its tensile properties, were stronger than those of EWHMa, and its tensile strength and elongation at break were over 5 MPa and 110%, respectively. The above results show that the ordering of the EW protein molecules was improved by UND, forming a crystal structure with partial α-helix characteristics. However, this structure was still water-soluble, and the molecular structure of the protein changed from α-helix to β-sheet structure only after high-temperature annealing, finally becoming a new material with significant improvement in mechanical properties: EWHMs.

### 2.12. Biodegradation In Vitro

In order to understand the different biodegradation rates of EWHMs towards different proteases, five protease aqueous solutions (140 U/mL) were used for the hydrolyzation of EWHMs at a constant temperature with shaking. Through the pre-experiment of enzymatic hydrolysis, we knew that the degradation rates of the two EWHMs were faster in trypsin and alkaline protease aqueous solutions but slower in the other three proteases. Therefore, the enzymatic hydrolysis experiment was divided into two test groups. One group was trypsin and alkaline protease solutions, and the residual rate of the sample was determined every 2 days. In the other group, the residual rate of the enzymatically hydrolyzed samples was determined after direct enzymatic hydrolysis by three kinds of proteases for 12 days. Two pre-treated EWGs were airtight oven-heated at 100 °C for 60 min or at 110 °C for 45 min to obtain EWHMa and EWHMb, respectively. The residual rates after enzymatic hydrolysis in trypsin and alkaline protease solution for 12 days are shown in Figure 9a. The enzymolysis residue of EWHM in the two protease solutions decreased with extended hydrolysis time, and the decreasing trend was basically the same for both solutions, but the degradation rate in the alkaline protease was significantly higher than that in trypsin. After the enzymatic hydrolysis for 12 days, almost all of the EWHMa was degraded, and only 11% of EWHMb remained, while 40% and 58% of the two kinds of EWHM were retained in the trypsin solution, respectively. The degradation of the two EWHMs in trypsin also has the same trend. That is, the higher the heat-treatment temperature of the EWG, the slower the degradation rate in alkaline protease and trypsin.

The experimental results of the other group of enzymatic hydrolysis in three protease solutions are shown in Figure 9b. The two EWHMs were shaken and hydrolyzed individually at constant temperature in neutral protease, pepsin, and papain aqueous solution for 12 days. It was observed that the degradation rates of EWHMa and EWHMb in papain solution were slightly faster than the other solutions, and the degradation residual rates were 44% and 72%, respectively. The degradation was much slower in the aqueous solution of the other two proteases (neutral protease and pepsin), and the range of enzymatic hydrolysis was only between 10% and 20%. Among them, the degradation rate was the slowest in pepsin aqueous solution, and after 12 days, the degradation rate of EWHMa was less than 3%, and that of EWHMb was less than 1%. 

Therefore, the enzymatic hydrolysis stability of water-soluble EWG was significantly improved after airtight oven heating at 100 °C for 60 min or at 110 °C for 45 min. The residual rate of EW after enzymatic hydrolysis for 12 days was pepsin > neutral protease > papain > trypsin > alkaline protease. 

### 2.13. L929 Growth and Proliferation on EWHMs

In order to determine the biocompatibility of the EW biomaterials developed in this study, the adhesion, growth, and proliferation of L929 fibroblasts on two EWHM samples were investigated. Figure 10a shows the growth of the L929 cells on the EWHM over 5 days. The cells grew normally on the control without EWHM on the fifth day, the L929 cells were fusiform and slender at both ends, and the nucleus could be observed after magnification. Because of the high cell density, some cells did not adhere to the wall and floated in the culture medium, forming clusters. The L929 cells in the two kinds of EWHM were fusiform, and the cytoplasm of the cell was also clearly visible. The whole cell was full and had a strong three-dimensional sense, and both ends appeared to be rooted in the EWHM. There was no significant difference in the growth of L929 cells between EWHMa and EWHMb (Figure 10b,c).

In this experiment, the cell viability of the two types of EWHM and the control group was measured using the CCK-8 method every day for five days. On the whole, the ABS_450_ value of the two sample groups was very close to that of the control group, and the L929 cells showed an almost linear proliferation trend. There was no significant difference between the two kinds of EWHM, and the cells adhered, grew, and proliferated on the two kinds of EWHM treated at different temperatures (Figure 10d). However, on the fifth day, the CCK-8 absorbance of the two groups of samples was about 5–8% lower than that of the control group, while there was almost no difference between the states of cell growth (Figure 10a–c). In fact, this slight difference may be because there was no EWHM in the control group, and there were EWHM in the sample group, which can adsorb the reactive dye during dyeing, and the dye cannot easily be washed out during determination, so the absorbance value will be decreased. In the future, it should be considered whether the sample breaks during dyeing and determination, the supernatant is taken after centrifugation, or the reactive dye is absorbed for the first time and then cleaned with the same amount of liquid. Then, the optical density is determined by combining the cleaning solution. This may reduce the error of the results between the control group and the EWHM group.

## 3. Discussion

The EW was steamed at 98 °C for 15 min to form EW hydrogel and then placed in a refrigerator at 5 °C and 60% RH to form a kind of EW hydrogel film through ultra-slow dehydration at a low temperature for two weeks [39]. However, its mechanical property was limited due to no post-treatment. Up until now, it has been rare to use EW as a single component and green processing into an EW hydrogel film with good mechanical properties, softness, and elasticity. Recently, authors have described a mechanically robust EW hydrogel scaffold with excellent biocompatibility by slow drying–heat treatment processing technology [37]. The hydrogel scaffold is water-insoluble, wet soft, and translucent and has a porous network structure with pore sizes ranging from 0.1 to 1.0 μm. Its tensile strength, elongation, and swelling rate in the wet state are 5.0 MPa, ~93%, and ~85%, respectively. In the present experiment, the EWHM obtained through UND followed by similar annealing is significantly superior to the former in terms of the above parameters. Why is there a significant difference in performance between the EW hydrogel scaffold and EWHM obtained through a combination technology of slow dehydration and subsequent annealing? The main reason is due to the different ways of slow dehydration. The former is the water-unstable EW hydrogel film formed by routine evaporation and drying of water in EW, while the latter is water-soluble and transparent EWG, dried by unidirectional dehydration of water in EW through nanopores. We used this same UND technology to make silk fibroin and PVA into three novel silk fibroin membranes [40], PVA hydrogel membrane [41], and PVA-xylosylated sericin composite hydrogel membrane [42] with strong mechanical properties and excellent stretching ability. 

The directional movement of water molecules through the nanopores of the film under gravity may cause EW protein molecules to accumulate in a directional manner and self-assemble on the nano-film [40]. The EW protein molecular chains formed are arranged in a more orderly manner, and the resulting EWG was water-soluble and translucent and had a porous network structure. After high-temperature annealing, the EWG became a transparent and yellow-reddish EWHM with good mechanical properties and elongation. This may be mainly caused by the following factors. First of all, high-temperature annealing leads to a Maillard reaction between reducing sugar and free amino acid residues in EWG protein, and this reaction not only causes complex cross-linking reactions between protein chains, but also turns EWHM yellow or light brown [38,43]. Second, EW protein molecules contain a large number of sulfhydryl groups [1]. The heat treatment induces the sulfhydryl–disulfide interchange between EW proteins, as reported by Mine et al. [44]. Third, the dehydration caused by annealing also causes the rearrangement of protein molecular chains or segments, resulting in further formation of a β-sheets structure, which significantly improves the mechanical properties of EWHM. Therefore, this joint technology of UND and annealing has the potential to be applied in the processing of new materials for more polymers.

## 4. Materials and Methods

### 4.1. EW Homogenization

The chicken egg in the experiment is the “Lanfei” brand produced by Dahe Egg products Co., Ltd. (Pudong District, Shanghai, China). After the eggs were broken, the EW was separated from the yolk with an EW separator. The mixture of EW thick and thin liquids was sheared at high speed (2000–5000 rpm) for 1–5 min. Then, the homogenized EW liquid was centrifuged at 8000 rpm for 20 min to remove the impurities and white floats, and the supernatant obtained was the homogenized EW for all experiments as follows. 

### 4.2. Preparation of Egg White Glass (EWG)

Before using the mold (Figure 1b) to prepare the EWG, the mold cup body (1) was covered with the nanopore filter membrane (2), then the fixing flange (3) was tightened. For a more detailed description of installation and use, readers can also refer to the literature report just published by the authors [38]. A small amount of distilled water was added through the sampling hole (4) to test whether the filter membrane leaked. Using the nanopore filter membrane, it was generally possible to intercept molecules of weight ≤ 1000 kDa or diameter ≤ 50 nm. Next, 5 mL of homogenized EW was added to the installed mold, and finally, the sampling hole was sealed with a rubber plug to make the mold airtight. The mold was placed on a horizontal mold rack at room temperature and dehydrated unidirectionally for up to tens of hours within certain temperature and relative humidity (RH) ranges. The EW liquid in the mold cup formed a fragile, transparent, and almost water-soluble EW glass.

### 4.3. Preparation of EW Hydrogel Membrane (EWHM)

This kind of transparent and fragile EWG must be annealed with heat treatment, that is, the EWG can be steamed, boiled, or heated for 5–120 min, in a bare or closed state, and a water-insoluble, wet, soft, and elastic EWHM with good mechanical properties can be obtained.

### 4.4. Determination of Mechanical Properties and Swelling Ratio

Before the mechanical properties were determined, the samples were pre-cut into rectangular strips (6.0 mm × 25 mm) and immersed in 37 °C water for 24 h. The tensile strength and elongation at break were then tested on the testing machine (INSTRON 3365, Glenview, IL, USA). According to the method just reported by the authors [37], the swelling ratio of these biosamples was detected and calculated. Ten samples were measured for each sample, and the average value and standard deviation (SD) were calculated.

### 4.5. Determination of In Vitro Degradation Rate and Cell Culture

According to the method described by Li et al. with some modification [38], these proteases (neutral protease, trypsin, pepsin, alkaline protease, and papain) were prepared into 140 U/mL protease stock solution with PBS. These filtered and sterilized protease solutions need to be dilute 10 times with PBS prior to use. The grouping and enzymolysis treatment of these EWHMs samples in the above five enzyme solutions, as well as the calculation of the residual rate of enzymolysis, were also carried out according to the methods previously reported by the authors [37].

The culture medium composition of mouse fibroblasts L929, culture methods and conditions, sample preparation, and grouping were carried out according to the methods in ref [38,40].

### 4.6. Thermostabilty, FTIR Spectrum, and XRD Pattern Analysis

Thermogravimetric (TG), derivative thermogravimetric (DTG), and differential scanning calorimetry (DSC), Fourier-transform infrared spectroscopy (FTIR) spectra, and X-ray diffraction (XRD) patterns for EWG and EWHM biosamples were measured according to the method and steps just reported by authors [37].

### 4.7. SEM Observation

Before scanning electron microscopy (SEM) observation, the samples were immersed in pure water for 24 h and frozen in liquid nitrogen. The cracked sections and surfaces were sprayed with gold for 60 s, and the apparent morphology was observed with a Hitachi S-4700 cold-field emission scanning electron microscope (Japan).

### 4.8. Statistical Methods

All the experimental data were analyzed with Origin 10 statistical software. Five replicants were prepared for each sample, and the calculated values are expressed in the form of mean values (±SD).

## 5. Conclusions

The EW hydrogel or EW membrane reported at present usually has poor mechanical properties, and it is difficult to apply it to biomaterials, especially medical biological materials, unless it is synthesized with other polymers. A homogeneous EW liquid was obtained by high-speed shearing and centrifugation. This green dehydration technology can dry not only the EW liquid but also form a kind of water-soluble, transparent, and fragile EWG with a dominant α-helix structure. Annealing can convert this EWG into a water-insoluble, translucent, and soft EW bioplastic with stronger mechanical properties. The annealing included watertight or airproof steaming, boiling, or oven heating at higher temperatures. The highest tensile strength of the EWHM was 5.84 MPa, the break at elongation was 50–110%, and the swelling ratio was 60–130%. Structure analysis showed that UND could increase the order in the arrangement of the protein molecules to form an EWG with a dominant α-helix structure. After annealing at a high temperature, the EWG transformed into an EWHM with a dominant β-sheet structure. Their 12-day residues in five proteases ranged from 1% to 99%, and the order was pepsin > neutral protease > papain > trypsin > alkaline protease. Mouse fibroblast L929 cells will adhere, grow, and proliferate well on these EWHMs. Therefore, this article combines the green and simple technology of UND and annealing to prepare an EWHM with not only solid mechanical properties but also good biocompatibility, which has potential application prospects in the fields of biomimetic and biomedical materials.

## Figures and Tables

**Figure 1 ijms-24-12661-f001:**
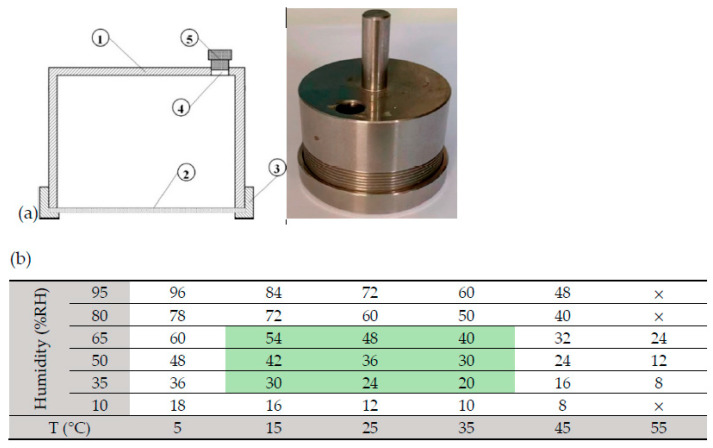
Structural diagram of UND mold and its photo (**a**) and effects of temperature (T) and humidity (RH) on the formation and formation time of EWG (**b**). (**a**) UND mold and its photo: ➀ film-forming cup body; ➁ filter film with nanopore; ➂ fixed flange, ➃ sampling hole; ➄ rubber plug. (**b**) The shaded area in the middle of Figure shows the time range within the temperature and humidity range for preparing the best performing EWG (the green part in the central area).

**Figure 2 ijms-24-12661-f002:**
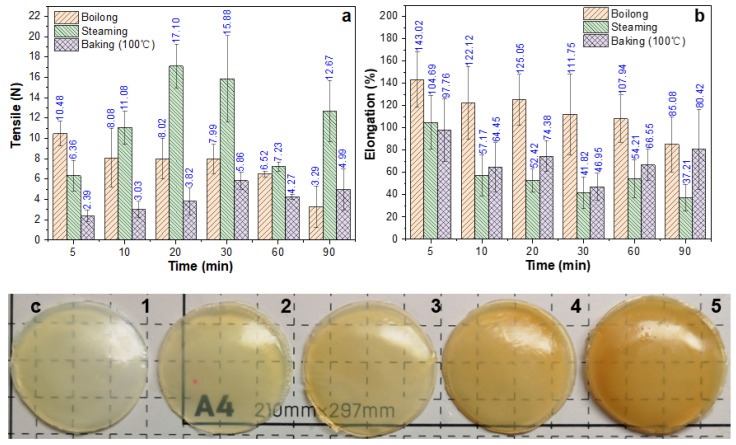
Effects of water boiling (**a**) and steaming and baking (**b**) at 100 °C on the tensile strengths, elongation rates, and appearance (**c**) of EWHM. **c**1–5, transparency and deepening color of EWHM formed by boiling EWG for 5, 10, 20, 30 and 60 min, respectively.

**Figure 3 ijms-24-12661-f003:**
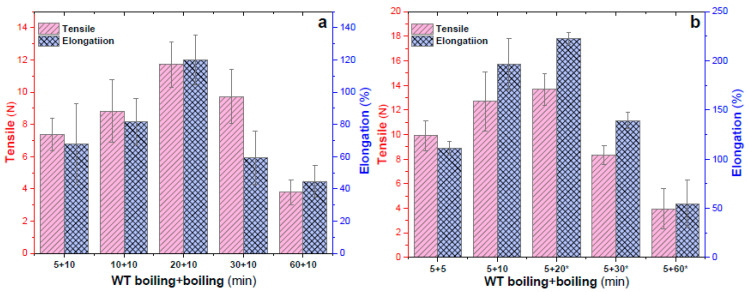
Effects of alternating water boiling (**a**) and watertight boiling (**b**) on mechanical properties and transparency; transparent but sticky surface, while all the other experimental groups are smooth, transparent, and non-sticky, indicating that if the exposed boiling time is too long, it will affect the transparency of the sample. WT: watertight. *, milky white EWHMs under this combined heat treatment.

**Figure 4 ijms-24-12661-f004:**
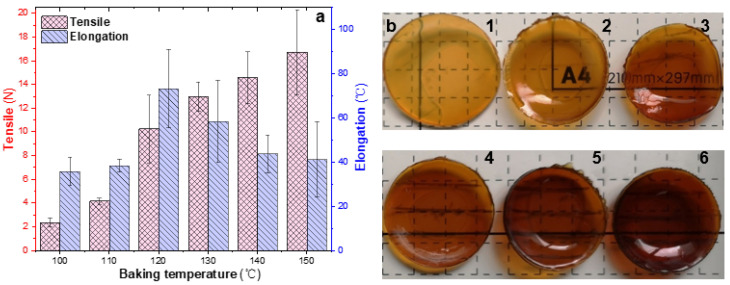
Effect of bake temperature on the mechanical properties (**a**) and appearance (**b**) of EWHM. The EWG samples were pre-treated by 10 min watertight boiling prior to baking treatment in oven. (**a**) Tensile and elongation; b1–6, baking 10 min at 100 °C, 110 °C, 120 °C, 130 °C, 140 °C, and 150 °C in oven, respectively.

**Figure 5 ijms-24-12661-f005:**
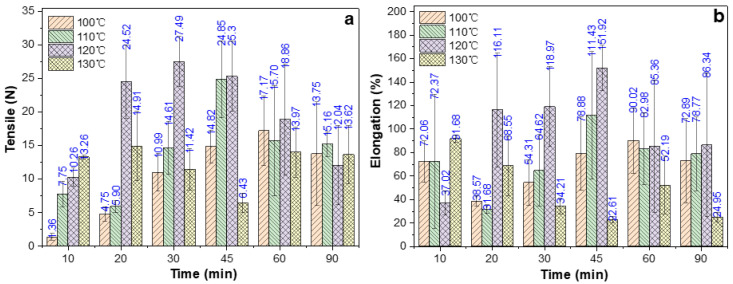
Effect of baking temperature and time on the tensile (**a**) and elongation (**b**) of EWHM formed by EWG (pre-treated with watertight boiling for 10 min).

**Figure 6 ijms-24-12661-f006:**
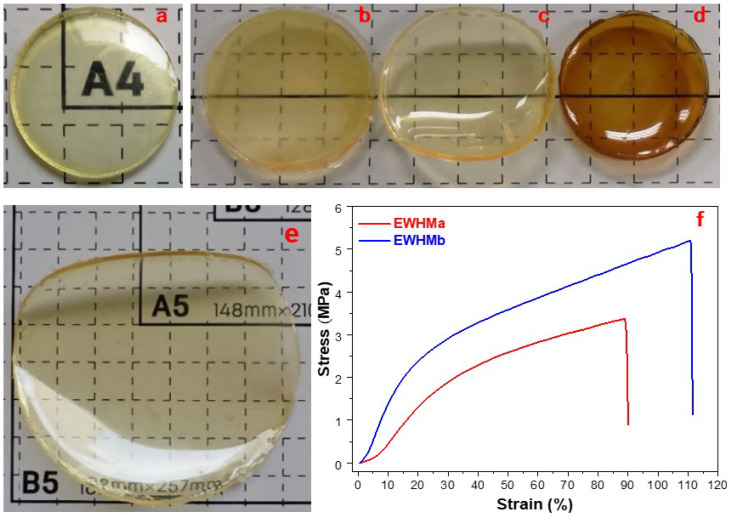
The appearance of EWG before annealing and stress and strain curves of EWHMs after annealing. (**a**) EWG; (**b**) EWHM (boiling for 10 min); (**c**) EWHM (watertight boiling for 10 min); (**d**) EWHM (baking for 30 min); (**e**) photograph of EWG with 80 mm diameter formed by heat treatment; (**f**) stress and strain curves of two kinds of EWHMs formed by heat treatment. EWG was first pre-treated by 10 min watertight boiling, and then two EWHMa and EWHMb were prepared by airtight baking at 100 °C for 60 min and at 110 °C for 45 min, respectively. The grid lines (ruler) in the figure are 10 mm × 10 mm.

**Figure 7 ijms-24-12661-f007:**
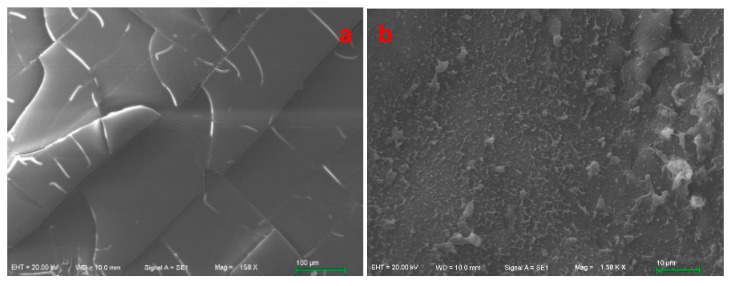
SEM photos of the surfa ce and cross-section of EWHMs. (**a**,**b**) The surface of UND-based EWG and EWHMa (airtight baking at 100 °C/60 min), respectively; (**c**,**d**) the cross-section of EWHMa at magnification ×200 and ×2000, respectively. The rulers are 100 μm and 10 μm in Figure 5a and 5b, respectively, and 200 μm and 20 μm in Figure 5c and 5d, respectively.

**Figure 8 ijms-24-12661-f008:**
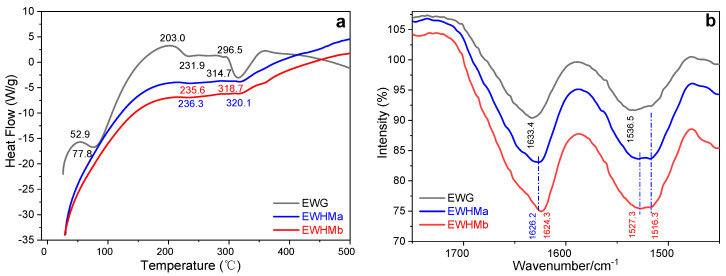
DSC patterns (**a**), FTIR spectra (**b**), and X-ray diffraction patterns (**c**) of EWG and EWHMs obtained after annealing. EWG prepared by UND was pre-treated by watertight boiling for 10 min, and then two EWHMs were prepared by airtight baking at 100 °C/60 min (EWHMa) and 110 °C/45 min (EWHMb), respectively.

**Figure 9 ijms-24-12661-f009:**
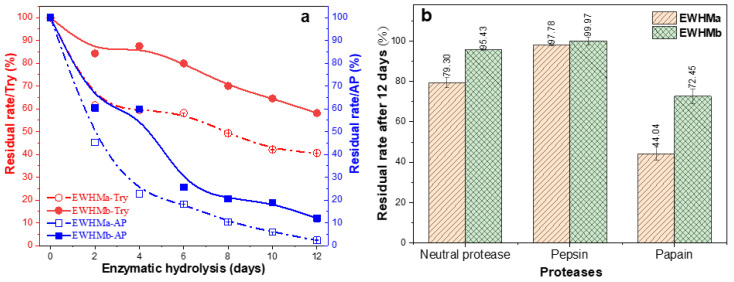
The degradation curves of two EWHMs against trypsin and alkaline protease (**a**) and neutral protease, pepsin, and papain (**b**) for 12 days.

**Figure 10 ijms-24-12661-f010:**
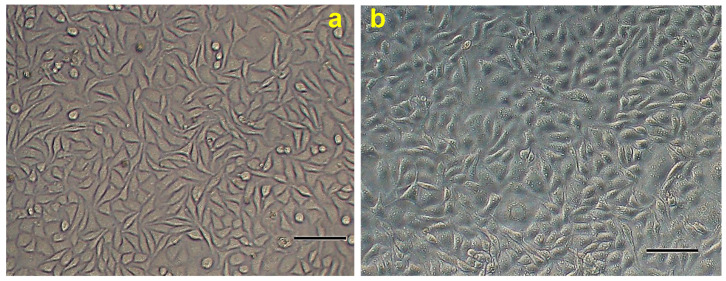
Growth and proliferation of L929 cells on two EWHMs. (**a**–**c**) L929 cells grew on 5th day on the culture plate without EWHM, with EWHMa and EWHMb, respectively; (**d**) the optical density changes of L929 cells grown on EWHMa and EWHMb on the 3rd, 4th, and 5th day were detected by CCK-8. The rulers in the picture are all 100 μm.

## Data Availability

Code and material; the datasets used and/or analyzed during the current study as well as analysis scripts are available from the corresponding author on reasonable request.

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
