# Peer review of "A Novel Mechanically Robust and Biodegradable Egg White Hydrogel Membrane by Combined Unidirectional Nanopore Dehydration and Annealing"

_ijms, 2023, doi:10.3390/ijms241612661_

Round 1

Reviewer 1 Report

The paper "A novel mechanically robust and biodegradable egg white hy-2 drogel membrane by combined unidirectional nanopore dehy-3 dration and annealing" is addreesed to processing egg white (EW).

In my opinion the topic of this paper might be interesting, but the aim of this paper is not clear enough to be understood. My general comment is that authors deeply revise the paper, first of all clarifying the aim of the paper.

Here below my detailed comments.

Introduction, line 42: the authors should specify if EW composition is the same independently of type of egg origin (chicken, goose, pigeon, etc...). If not, they should indicate that the work is addressed to EW from chicken eggs.

Results, line 96: The figure and figure  legend are not clear. Please explain what are referred to the numeber reported in figure 1a . Please explain which is the meaning of the green square in the middle of the Table of Figure 1b.

Results, line 118: The authors should evaluate and explain why the different heating methods affects EW characteristics. 

Results, Figure 2: The legend of Figure 2 is wrong because Figure 2a reports the tensile strenghts and Figure 2b reports elongation at breaks values of EW treated with the different heating time and methods tested. Figure 2c shows the appearance of EW treated with different heating methods, but is not clear what are the numeber referred to. The authors should correct and the Figure legend should be more synthetic and clear.

Results, line 157: Please indicate which panel of figure 3 you are referring here. 

Results, line 192, Figure 3: Figure 3 in my opinion is very difficult to be followed because it unifies too many  different figures. My suggestion is to separate in another Figure at least Figure 3 a and b.

Results, line 214: The authors should better investigate and explain the causes of color change. 

Materials and methods, line 396: In my opinion the EW homogenization step should include a filtration step through 0.22 micron membrane in order to get the EW more reproducible.  Filtration through 0.22 micron membrane coould be profitable also to withdrawn contaminant microorganisms.  

English language is clear. It requires minor revision. 

Author Response

Dear Editor,

Based on the opinions of the three reviewers, we have made our best efforts to make revisions, and as a result, we have added the discussion section.

It should be pointed out here that the third reviewer's opinion is particularly difficult to answer, such as spraying gold operation before observing SEM samples, which is a routine experimental step. As a result, the reviewer requested me to measure the thickness of the surface spraying gold on the sample.

The revised parts or sentences in the manuscript are marked in red font.

Thank you very much!

July 27, 2023

Reviewer 2 Report

The field of applications of membranes obtained from egg white is not new, but in this article it is adequately presented. The authors appropriately described the methodology for obtaining the cell membranes as well as a wide range of experiments to characterize these membranes as completely as possible. Their applicability is, as described, very wide.

My suggestions for improvement are the following:

1. A wider presentation of data from specialized literature regarding medical applications, with recent examples. In Introduction sectiun should be precisely prezented also some economic aspect related cu the subject of egg white hydrogel membranes. 

2. Highlighting the compositional problems of raw materials from different suppliers and how this aspect can be overcome.

3. The authors are asked to present which stability problems they had in mind, and if these studies are necessary, in which situations. This last aspect must be the subject of the  Results /Discussion and  Conclusions sections.

4. Also in Conclusions area more references to future studies regarding applicability directly in the medical field are desirable. What could be taken into account more precisely in order to be able to finalize a final product based on egg white.

6.  A standardization of the resolution of the fichus and graifces throughout the manuscript is necessary.

7.The list of bibliographic references must be re-verified in order to precisely comply with the requirements of the Authors' Instructions.

Based on my own proficiency in English, an extensive editing of English language required this manuscript.

Author Response

(The authors gave the same response as above.)

Reviewer 3 Report

The authors presented a new method for producing biodegradable hydrogels from white eggs. The subject is interesting. However, the manuscript presents some major issues that have to be addressed.

  • The introduction is non-update; more recent papers must be added, maybe including other biobased and biodegradable hydrogels examples.
  • The "unidirectional nanopore dehydration" method, first mentioned in the introduction section line 72, is not clearly described. The authors should explain in more detail the process.
  • In the 2.1 section, I believe it could be better to add a picture of the apparatus used along with the obtained dried structure rather than just a scheme.
  • Section 2.2 . it is unclear why the author chose a dehydration method and then two "annealing " processes in wet conditions. In this case, was it not enough just put the white egg at 100°C without drying it? Or if the drying process is not reversible, could the authors explain way from a chemical point of view?
  • The effect of the temperature on the proteins should be discussed in more detail. 
  • Line 113 "Why was it not effective? Can the authors comment more on the elongation at break results? Why does the oven treatment improve the elongation at break?" . Why was it not effective? can the authors comment more on the elongation at break results? Why, in this case, the oven treatment improved the elongation at break? A more detailed discussion should be added.
  • Fig. 2 The authors should add the results of the untreated (just dried) EWG in the graph as a comparison. Moreover, the authors should present the values of the Yong moduli obtained by each sample rather than the tensile force and put the stress-strain curves in the supporting information.
  • Line 134 "watertight boil" What does it mean?

-Line 164 "However, the higher the treatment temperature was, the worse the swelling ratio was." this phrase should be re-written

- Alle the tests presented need to include comparison with the literature. Are those values comparable to the ones obtained before with other biobased materials / biodegradable materials? Are these hydrogels competitive? That information will also help to highlight the value of this work.

- Line 207 -Can the authors provide a chemical explanation for this? How can a water-soluble sample immersed in hot water not dissolve but become insoluble?

-Line 215 A more detailed description of the Maillard reaction is needed

- Line 225 it is mentioned a material testing machine; please clarify which type.

- Line 249 - The authors used the term "bioglass" which I believe is incorrect.

-Paragraph 2.8, can the authors provide a more detailed discussion on the different structures observed by suggesting possible explanations? 

-Paragraph 2.9 Can the authors add some literature references for that TGA observation? 

-Line 321 - check the phrase

-Paragraph 2.12 and 2.13 Since the authors seam to have already evaluated those results in another article concerning the preparation of egg white hydrogel scaffold, why the present results are not compared to the previous work? I also think it is better to add other materials based on proteins and used for hydrogels as comparisons.

- Paragraph 3.4 Please describe the method and add the ASTM or ISO or some standard followed to confirm the validity of the results

-Paragraphs 3.5 and 3.6 Please describe the method in detail. A self-citation is not enough.

- Paragraph 3.7 spraying with gold? Can the author provide the thickness of the Au layer deposition?

The text needs to be double-checked.

Author Response

(The authors gave the same response as above.)

Round 2

Reviewer 1 Report

Dear authors,

I appreciate your effort in following the reviewer suggestions.

The paper il enough clear in the present form. 

Author Response

thank you very much

Reviewer 2 Report

The manuscript has been sufficiently improved, and after fine revisions for English language I could propose it for publication.

Based on my own proficiency in English, I am qualified and able to assess that English language required minor editing corrections.

Author Response

Thank you very much

Reviewer 3 Report

The renewed version of the manuscript is slightly improved from the last version. However, there are still some significant issues that have to be addressed:

1)The comparison of the results of this work with the literature is still missing. There should be added comparison with other hydrogel membranes used for biomedical applications, not only a brief phrase on the consistency of these results with work done by the author on the same starting material. I mean, for example, the heat resistances of those materials, observed by TGA, are comparable with the one reported with some other hydrogels membrane used for biomedical applications? Like made from gelatin, starch, chitosan, etc...? Same for the other tests performed by authors.

2) It is a shame that the authors do not have the time to investigate better the possible causes of the different SEM structures obtained. This could have added an interesting discussion to the present article.

Anyway, I asked the question on the gold layer thickness because it is really unusual to find “spry gold” and not “deposited gold”. In fact, for SEM, it is more common to have a gold deposition (vacuum assisted), and with this technique, you can control the thickness of the gold up to the nm scale. I think it is really important to know the thickness of the gold to evaluate the observed surfaces better. 

3) I still believe that the young moduli values have to be added in the discussion section, otherwise it is really impossible to compare the results presented here with other works reported in the literature. Moreover, the young modulus provides important information on the elasticity of the material, and a discussion on that is totally missing in the article.

4) Line 181: "However, the higher the treatment temperature was, the worse the elongation ratio was." Worse? what does it mean ? Maybe the authors would consider to re-write this sentence again. I may suggest: " Interestingly, the increase of the temperature of the process leads to a reduction of the elongation at break"

The English level of this manuscript has still to be improved.

Author Response

Dear Editor,

Based on your opinions, we have made our best efforts to make revisions.

The revised parts or sentences in the manuscript are marked in red font.

Thank you very much!

Sept 6, 2023

Round 3

Reviewer 3 Report

No comment.

Average